# Is Approval Voting Optimal Given Approval Votes?

**Ariel D. Procaccia**
Computer Science Department
Carnegie Mellon University
arielpro@cs.cmu.edu

**Nisarg Shah**
Computer Science Department
Carnegie Mellon University
nkshah@cs.cmu.edu

## Abstract

Some crowdsourcing platforms ask workers to express their opinions by approving a set of $k$ good alternatives. It seems that the only reasonable way to aggregate these *k-approval votes* is the *approval voting rule*, which simply counts the number of times each alternative was approved. We challenge this assertion by proposing a probabilistic framework of noisy voting, and asking whether approval voting yields an alternative that is most likely to be the best alternative, given $k$-approval votes. While the answer is generally positive, our theoretical and empirical results call attention to situations where approval voting is suboptimal.

## 1   Introduction

It is surely no surprise to the reader that modern machine learning algorithms thrive on large amounts of data — preferably labeled. Online labor markets, such as *Amazon Mechanical Turk* (www.mturk.com), have become a popular way to obtain labeled data, as they harness the power of a large number of human workers, and offer significantly lower costs compared to expert opinions. But this low-cost, large-scale data may require compromising quality: the workers are often unqualified or unwilling to make an effort, leading to a high level of noise in their submitted labels.

To overcome this issue, it is common to hire multiple workers for the same task, and aggregate their noisy opinions to find more accurate labels. For example, TurKit [17] is a toolkit for creating and managing crowdsourcing tasks on Mechanical Turk. For our purposes its most important aspect is that it implements plurality voting: among available alternatives (e.g., possible labels), workers report the best alternative in their opinion, and the alternative that receives the most votes is selected.

More generally, workers may be asked to report the $k$ best alternatives in their opinion; such a vote is known as a $k$-approval vote. This has an advantage over plurality (1-approval) in noisy situations where a worker may not be able to pinpoint the best alternative accurately, but can recognize that it is among the top $k$ alternatives [23].[1] At the same time, $k$-approval votes, even for $k > 1$, are much easier to elicit than, say, rankings of the alternatives, not to mention full utility functions. For example, EteRNA [16] — a citizen science game whose goal is to design RNA molecules that fold into stable structures — uses $8$-approval voting on submitted designs, that is, each player approves up to $8$ favorite designs; the designs that received the largest number of approval votes are selected for synthesis in the lab.

So, the elicitation of $k$-approval votes is common practice and has significant advantages. And it may seem that the only reasonable way to aggregate these votes, once collected, is via the *approval voting* rule, that is, tally the number of approvals for each alternative, and select the most approved one.[2] But is it? In other words, do the $k$-approval votes contain useful information that can lead to

significantly better outcomes, and is ignored by approval voting? Or is approval voting an (almost) optimal method for aggregating $k$-approval votes?

**Our Approach.** We study the foregoing questions within the *maximum likelihood estimation* (MLE) framework of social choice theory, which posits the existence of an underlying ground truth that provides an objective comparison of the alternatives. From this viewpoint, the votes are noisy estimates of the ground truth. The optimal rule then selects the alternative that is most likely to be the best alternative given the votes. This framework has recently received attention from the machine learning community [18, 3, 2, 4, 21], in part due to its applications to crowdsourcing domains [20, 21, 9], where, indeed, there is a ground truth, and individual votes are objective.

In more detail, in our model there exists a ground truth ranking over the alternatives, and each *voter* holds an opinion, which is another ranking that is a noisy estimate of the ground truth ranking. The opinions are drawn i.i.d. from the popular Mallows model [19], which is parametrized by the ground truth ranking, a noise parameter $\varphi \in [0, 1]$, and a distance metric $d$ over the space of rankings. We use five well-studied distance metrics: the Kendall tau (KT) distance, the (Spearman) footrule distance, the maximum displacement distance, the Cayley distance, and the Hamming distance.

When required to submit a $k$-approval vote, a voter simply approves the top $k$ alternatives in his opinion. Given the votes, an alternative $a$ is the *maximum likelihood estimate* (MLE) for the best alternative if the votes are most likely generated by a ranking that puts $a$ first.

We can now reformulate our question in slightly more technical terms:

> *Is approval voting (almost) a maximum likelihood estimator for the best alternative, given votes drawn from the Mallows model? How does the answer depend on the noise parameter $\phi$ and the distance metric $d$?*

**Our results.** Our first result (Theorem 1) shows that under the Mallows model, the set of winners according to approval voting coincides with the set of MLE best alternatives under the Kendall tau distance, but under the other four distances there may exist approval winners that are not MLE best alternatives. Our next result (Theorem 2) confirms the intuition that the suboptimality of approval voting stems from the information that is being discarded: when only a single alternative is approved or disapproved in each vote, approval voting — which now utilizes all the information that can be gleaned from the anonymous votes — is optimal under mild conditions.

Going back to the general case of $k$-approval votes, we show (Theorem 3) that even under the four distances for which approval voting is suboptimal, a weaker statement holds: in cases with very high or very low noise, every MLE best alternative is an approval winner (but some approval winners may not be MLE best alternatives). And our experiments, using real data, show that the accuracy of approval voting is usually quite close to that of the MLE in pinpointing the best alternative.

We conclude that approval voting is a good way of aggregating $k$-approval votes in most situations. But our work demonstrates that, perhaps surprisingly, approval voting may be suboptimal, and, in situations where a high degree of accuracy is required, exact computation of the MLE best alternative is an option worth considering. We discuss our conclusions in more detail in Section 6.

## 2 Model

Let $[t] \triangleq \{1, \ldots, t\}$. Denote the set of *alternatives* by $A$, and let $|A| = m$. We use $\mathcal{L}(A)$ to denote the set of rankings (total orders) of the alternatives in $A$. For a ranking $\sigma \in \mathcal{L}(A)$, let $\sigma(i)$ denote the alternative occupying position $i$ in $\sigma$, and let $\sigma^{-1}(a)$ denote the rank (position) of alternative $a$ in $\sigma$. With a slight abuse of notation, let $\sigma([t]) \triangleq \{a \in A | \sigma^{-1}(a) \in [t]\}$. We use $\sigma_{a \leftrightarrow b}$ to denote the ranking obtained by swapping the positions of alternatives $a$ and $b$ in $\sigma$. We assume that there exists an unknown *true ranking* of the alternatives (the ground truth), denoted $\sigma^* \in \mathcal{L}(A)$. We also make the standard assumption of a uniform prior over the true ranking.

---

framework of approval voting has been studied extensively, both from the axiomatic point of view [7, 8, 13, 22, 1], and the game-theoretic point of view [14, 12, 6]. However, even under this framework it is a standard assumption that votes are tallied by counting the number of times each alternative is approved, which is why we simply refer to the aggregation rule under consideration as approval voting.

Let $N = \{1, \ldots, n\}$ denote the set of *voters*. Each voter $i$ has an *opinion*, denoted $\pi_i \in \mathcal{L}(A)$, which is a noisy estimate of the true ranking $\sigma^*$; the collection of opinions — the (opinion) *profile* — is denoted $\pi$. Fix $k \in [m]$. A *$k$-approval vote* is a collection of $k$ alternatives approved by a voter. When asked to submit a $k$-approval vote, voter $i$ simply submits the *vote* $V_i = \pi_i([k])$, which is the set of alternatives at the top $k$ positions in his opinion. The collection of all votes is called the *vote profile*, and denoted $V = \{V_i\}_{i \in [n]}$. For a ranking $\sigma$ and a $k$-approval vote $v$, we say that $v$ is generated from $\sigma$, denoted $\sigma \to_k v$ (or $\sigma \to v$ when the value of $k$ is clear from the context), if $v = \sigma([k])$. More generally, for an opinion profile $\pi$ and a vote profile $V$, we say $\pi \to_k V$ (or $\pi \to V$) if $\pi_i \to_k V_i$ for every $i \in [n]$.

Let $\mathcal{A}_k = \{A_k \subseteq A || A_k | = k\}$ denote the set of all subsets of $A$ of size $k$. A *voting rule* operating on $k$-approval votes is a function $(\mathcal{A}_k)^n \to A$ that returns a *winning alternative* given the votes.[3] In particular, let us define the approval score of an alternative $a$, denoted $\mathrm{SC}^{\mathrm{APP}}(a)$, as the number of voters that approve $a$. Then, *approval voting* simply chooses an alternative with the greatest approval score. Note that we do not break ties. Instead, we talk about the *set of approval winners*.

Following the standard social choice literature, we model the opinion of each voter as being drawn i.i.d. from an underlying *noise model*. A noise model describes the probability of drawing an opinion $\sigma$ given the true ranking $\sigma^*$, denoted $\Pr[\sigma|\sigma^*]$. We say that a noise model is *neutral* if the labels of the alternatives do not matter, i.e., renaming alternatives in the true ranking $\sigma$ and in the opinion $\sigma^*$, in the same fashion, keeps $\Pr[\sigma|\sigma^*]$ intact. A popular noise model is the *Mallows model* [19], under which $\Pr[\sigma|\sigma^*] = \varphi^{d(\sigma,\sigma^*)}/Z_\varphi^m$. Here, $d$ is a distance metric over the space of rankings. Parameter $\varphi \in [0, 1]$ governs the noise level; $\varphi = 0$ implies that the true ranking is generated with probability 1, and $\varphi = 1$ implies the uniform distribution. $Z_\varphi^m$ is the normalization constant, which is independent of the true ranking $\sigma^*$ given that distance $d$ is *neutral*, i.e., renaming alternatives in the same fashion in two rankings does not change the distance between them. Below, we review five popular distances used in the social choice literature; they are all neutral.

- *The Kendall tau (KT) distance*, denoted $d_{KT}$, measures the number of pairs of alternatives over which two rankings disagree. Equivalently, it is the number of swaps required by bubble sort to convert one ranking into another.

- *The (Spearman) footrule (FR) distance*, denoted $d_{FR}$, measures the total displacement (absolute difference between positions) of all alternatives in two rankings.

- *The Maximum Displacement (MD) distance*, denoted $d_{MD}$, measures the maximum of the displacements of all alternatives between two rankings.

- *The Cayley (CY) distance*, denoted $d_{CY}$, measures the minimum number of swaps (not necessarily of adjacent alternatives) required to convert one ranking into another.

- *The Hamming (HM) distance*, denoted $d_{HM}$, measures the number of positions in which two rankings place different alternatives.

Since opinions are drawn independently, the probability of a profile $\pi$ given the true ranking $\sigma^*$ is $\Pr[\pi|\sigma^*] = \prod_{i=1}^n \Pr[\pi_i|\sigma^*] \propto \varphi^{d(\pi,\sigma^*)}$, where $d(\pi, \sigma^*) = \sum_{i=1}^n d(\pi_i, \sigma^*)$. Once we fix the noise model, for a fixed $k$ we can derive the probability of observing a given $k$-approval vote $v$: $\Pr[v|\sigma^*] = \sum_{\sigma \in \mathcal{L}(A):\sigma \to v} \Pr[\sigma|\sigma^*]$. Then, the probability of drawing a given vote profile $V$ is $\Pr[V|\sigma^*] = \prod_{i=1}^n \Pr[V_i|\sigma^*]$. Alternatively, this can also be expressed as $\Pr[V|\sigma^*] = \sum_{\pi \in \mathcal{L}(A)^n:\pi \to V} \Pr[\pi|\sigma^*]$. Hereinafter, we omit the domains $\mathcal{L}(A)^n$ for $\pi$ and $\mathcal{L}(A)$ for $\sigma^*$ when they are clear from the context.

Finally, given the vote profile $V$ the likelihood of an alternative $a$ being the best alternative in the true ranking $\sigma^*$ is proportional to (via Bayes' rule) $\Pr[V|\sigma^*(1) = a] = \sum_{\sigma^*:\sigma^*(1)=a} \Pr[V|\sigma^*]$. Using the two expressions derived earlier for $\Pr[V|\sigma^*]$, and ignoring the normalization constant $Z_\varphi^m$ from the probabilities, we define the likelihood function of $a$ given votes $V$ as

$$L(V, a) \triangleq \sum_{\sigma^*:\sigma^*(1)=a} \sum_{\pi:\pi \to V} \varphi^{d(\pi,\sigma^*)} = \sum_{\sigma^*:\sigma^*(1)=a} \prod_{i=1}^n \left[ \sum_{\pi_i:\pi_i \to V_i} \varphi^{d(\pi_i,\sigma^*)} \right]. \qquad (1)$$

The *maximum likelihood estimate* (MLE) for the best alternative is given by $\arg\max_{a \in A} L(V, a)$. Again, we do not break ties; we study the *set of MLE best alternatives*.

# 3 Optimal Voting Rules

At first glance, it seems natural to use approval voting (that is, returning the alternative that is approved by the largest number of voters) given $k$-approval votes. However, consider the following example with 4 alternatives ($A = \{a, b, c, d\}$) and 5 voters providing 2-approval votes:

$$V_1 = \{b, c\}, \ V_2 = \{b, c\}, \ V_3 = \{a, d\}, \ V_4 = \{a, b\}, \ V_5 = \{a, c\}. \tag{2}$$

Notice that alternatives $a$, $b$, and $c$ receive 3 approvals each, while alternative $d$ receives only a single approval. Approval voting may return any alternative other than alternative $d$. But is that always optimal? In particular, while alternatives $b$ and $c$ are symmetric, alternative $a$ is qualitatively different due to different alternatives being approved along with $a$. This indicates that under certain conditions, it is possible that not all three alternatives are MLE for the best alternative. Our first result shows that this is indeed the case under three of the distance functions listed above, and a similar example works for a fourth. However, surprisingly, under the Kendall tau distance the MLE best alternatives are exactly the approval winners, and hence are polynomial-time computable, which stands in sharp contrast to the NP-hardness of computing them given *rankings* [5].

**Theorem 1.** *The following statements hold for aggregating $k$-approval votes using approval voting.*

1. *Under the Mallows model with a fixed distance $d \in \{d_{MD}, d_{CY}, d_{HM}, d_{FR}\}$, there exist a vote profile $V$ with at most six 2-approval votes over at most five alternatives, and a choice for the Mallows parameter $\varphi$, such that not all approval winners are MLE best alternatives.*

2. *Under the Mallows model with the distance $d = d_{KT}$, the set of MLE best alternatives coincides with the set of approval winners, for all vote profiles $V$ and all values of the Mallows parameter $\varphi \in (0, 1)$.*

*Proof.* For the Mallows model with $d \in \{d_{MD}, d_{CY}, d_{HM}\}$ and any $\varphi \in (0, 1)$, the profile from Equation (2) is a counterexample: alternatives $b$ and $c$ are MLE best alternatives, but $a$ is not. For the Mallows model with $d = d_{FR}$, we could not find a counter example with 4 alternatives; computer-based simulations generated the following counterexample with 5 alternatives that works for any $\varphi \in (0, 1)$: $V_1 = V_2 = \{a, b\}$, $V_3 = V_4 = \{c, d\}$, $V_5 = \{a, e\}$, and $V_6 = \{b, c\}$. Here, alternatives $a$, $b$, and $c$ have the highest approval score of 3. However, alternative $b$ has a strictly lower likelihood of being the best alternative than alternative $a$, and hence is not an MLE best alternative. The calculation verifying these counterexamples is presented in the online appendix (specifically, Appendix A).

In contrast, for the Kendall tau distance, we show that all approval winners are MLE best alternatives, and vice-versa. We begin by simplifying the likelihood function $L(V, a)$ from Equation (1) for the special case of the Mallows model with the Kendall tau distance. In this case, it is well known that the normalization constant satisfies $Z_\varphi^m = \prod_{j=1}^m T_\varphi^j$, where $T_\varphi^j = \sum_{i=0}^{j-1} \varphi^i$. Consider a ranking $\pi_i$ such that $\pi_i \to V_i$. We can decompose $d_{KT}(\pi_i, \sigma^*)$ into three types of pairwise mismatches: i) $d_1(\pi_i, \sigma^*)$: The mismatches over pairs $(b, c)$ where $b \in V_i$ and $c \in A \setminus V_i$, or vice-versa; ii) $d_2(\pi_i, \sigma^*)$: The mismatches over pairs $(b, c)$ where $b, c \in V_i$; and iii) $d_3(\pi_i, \sigma^*)$: The mismatches over pairs $(b, c)$ where $b, c \in A \setminus V_i$.

Note that every ranking $\pi_i$ that satisfies $\pi_i \to V_i$ has identical mismatches of type 1. Let us denote the number of such mismatches by $d_{KT}(V_i, \sigma^*)$. Also, notice that $d_2(\pi_i, \sigma^*) = d_{KT}(\pi_i|_{V_i}, \sigma^*|_{V_i})$, where $\sigma|_S$ denotes the ranking of alternatives in $S \subseteq A$ dictated by $\sigma$. Similarly, $d_3(\pi_i, \sigma^*) = d_{KT}(\pi_i|_{A \setminus V_i}, \sigma^*|_{A \setminus V_i})$. Now, in the expression for the likelihood function $L(V, a)$,

$$L(V, a) = \sum_{\sigma^*: \sigma^*(1)=a} \prod_{i=1}^n \sum_{\pi_i: \pi_i \to V} \varphi^{d_{KT}(V_i, \sigma^*) + d_{KT}(\pi_i|_{V_i}, \sigma^*|_{V_i}) + d_{KT}(\pi_i|_{A \setminus V_i}, \sigma^*|_{A \setminus V_i})}$$

$$= \sum_{\sigma^*: \sigma^*(1)=a} \prod_{i=1}^n \varphi^{d_{KT}(V_i, \sigma^*)} \left[ \sum_{\pi_i^1 \in \mathcal{L}(V_i)} \varphi^{d_{KT}(\pi_i^1, \sigma^*|_{V_i})} \right] \cdot \left[ \sum_{\pi_i^2 \in \mathcal{L}(A \setminus V_i)} \varphi^{d_{KT}(\pi_i^2, \sigma^*|_{A \setminus V_i})} \right]$$

$$= \sum_{\sigma^*: \sigma^*(1)=a} \prod_{i=1}^n \varphi^{d_{KT}(V_i, \sigma^*)} \cdot Z_\varphi^k \cdot Z_\varphi^{m-k} \propto \sum_{\sigma^*: \sigma^*(1)=a} \varphi^{d_{KT}(V, \sigma^*)} \triangleq \widehat{L}(V, a).$$

The second equality follows because every ranking $\pi_i$ that satisfies $\pi_i \to V$ can be generated by picking rankings $\pi_i^1 \in \mathcal{L}(V_i)$ and $\pi_i^2 \in \mathcal{L}(A \setminus V_i)$, and concatenating them. The third equality follows from the definition of the normalization constant in the Mallows model. Finally, we denote $d_{KT}(V, \sigma^*) \triangleq \sum_{i=1}^n d_{KT}(V_i, \sigma^*)$. It follows that maximizing $L(V, a)$ amounts to maximizing $\widehat{L}(V, a)$. Note that $d_{KT}(V, \sigma^*)$ counts the number of times alternative $a$ is approved while alternative $b$ is not for all $a, b \in A$ with $b \succ_{\sigma^*} a$. That is, let $n^V(a, -b) \triangleq |\{i \in [n] | a \in V_i \wedge b \notin V_i\}|$. Then, $d_{KT}(V, \sigma^*) = \sum_{a,b \in A : b \succ_{\sigma^*} a} n^V(a, -b)$. Also, note that for alternatives $c, d \in A$, we have $\mathrm{SC}^{\mathrm{APP}}(c) - \mathrm{SC}^{\mathrm{APP}}(d) = n^V(c, -d) - n^V(d, -c)$.

Next, we show that $\widehat{L}(V, a)$ is a monotonically increasing function of $\mathrm{SC}^{\mathrm{APP}}(a)$. Equivalently, $\widehat{L}(V, a) \geq \widehat{L}(V, b)$ if and only if $\mathrm{SC}^{\mathrm{APP}}(a) \geq \mathrm{SC}^{\mathrm{APP}}(b)$. Fix $a, b \in A$. Consider the bijection between the sets of rankings placing $a$ and $b$ first, which simply swaps $a$ and $b$ ($\sigma \leftrightarrow \sigma_{a \leftrightarrow b}$). Then,

$$\widehat{L}(V, a) - \widehat{L}(V, b) = \sum_{\sigma^* : \sigma^*(1) = a} \varphi^{d_{KT}(V, \sigma^*)} - \varphi^{d_{KT}(V, \sigma^*_{a \leftrightarrow b})}. \tag{3}$$

Fix $\sigma^*$ such that $\sigma^*(1) = a$. Note that $\sigma^*_{a \leftrightarrow b}(1) = b$. Let $C$ denote the set of alternatives positioned between $a$ and $b$ in $\sigma^*$ (equivalently, in $\sigma^*_{a \leftrightarrow b}$). Now, $\sigma^*$ and $\sigma^*_{a \leftrightarrow b}$ have identical disagreements with $V$ on a pair of alternatives $(x, y)$ unless i) one of $x$ and $y$ belongs to $\{a, b\}$, and ii) the other belongs to $C \cup \{a, b\}$. Thus, the difference of disagreements of $\sigma^*$ and $\sigma^*_{a \leftrightarrow b}$ with $V$ on such pairs is

$$d_{KT}(V, \sigma^*) - d_{KT}(V, \sigma^*_{a \leftrightarrow b})$$
$$= \left[ n^V(b, -a) - n^V(a, -b) \right] + \sum_{c \in C} [n^V(c, -a) + n^V(b, -c) - n^V(c, -b) - n^V(a, -c)]$$
$$= (|C| + 1) \cdot \left( \mathrm{SC}^{\mathrm{APP}}(b) - \mathrm{SC}^{\mathrm{APP}}(a) \right).$$

Thus, $\mathrm{SC}^{\mathrm{APP}}(a) = \mathrm{SC}^{\mathrm{APP}}(b)$ implies $d_{KT}(V, \sigma^*) = d_{KT}(V, \sigma^*_{a \leftrightarrow b})$ (and thus, $\widehat{L}(V, a) = \widehat{L}(V, b)$), and $\mathrm{SC}^{\mathrm{APP}}(a) > \mathrm{SC}^{\mathrm{APP}}(b)$ implies $d_{KT}(V, \sigma^*) < d_{KT}(V, \sigma^*_{a \leftrightarrow b})$ (and thus, $\widehat{L}(V, a) > \widehat{L}(V, b)$). ∎

Suboptimality of approval voting for distances other than the KT distance stems from the fact that in counting the number of approvals for a given alternative, one discards information regarding other alternatives approved along with the given alternative in various votes. However, no such information is discarded when only one alternative is approved (or not approved) in each vote. That is, given plurality ($k = 1$) or veto ($k = m - 1$) votes, approval voting should be optimal, not only for the Mallows model but for any reasonable noise model. The next result formalizes this intuition.

**Theorem 2.** *Under a neutral noise model, the set of MLE best alternatives coincides with the set of approval winners*

1. *given plurality votes, if $p_1 > p_i > 0, \forall i \in \{2, \ldots, m\}$, where $p_i$ is the probability of the alternative in position $i$ in the true ranking appearing in the first position in a sample, or*

2. *given veto votes, if $0 < q_1 < q_i, \forall i \in \{2, \ldots, m\}$, where $q_i$ is the probability of the alternative in position $i$ in the true ranking appearing in the last position in a sample.*

*Proof.* We show the proof for plurality votes. The case of veto votes is symmetric: in every vote, instead of a single approved alternative, we have a single alternative that is not approved. Note that the probability $p_i$ is independent of the true ranking $\sigma^*$ due to the neutrality of the noise model.

Consider a plurality vote profile $V$ and an alternative $a$. Let $T = \{\sigma^* \in \mathcal{L}(A) | \sigma^*(1) = a\}$. The likelihood function for $a$ is given by $L(V, a) = \sum_{\sigma^* \in T} \Pr[V | \sigma^*]$. Under every $\sigma^* \in T$, the contribution of the $\mathrm{SC}^{\mathrm{APP}}(a)$ plurality votes for $a$ to the product $\Pr[V | \sigma^*] = \prod_{i=1}^n \Pr[V_i | \sigma^*]$ is $(p_1)^{\mathrm{SC}^{\mathrm{APP}}(a)}$. Note that the alternatives in $A \setminus \{a\}$ are distributed among positions in $\{2, \ldots, m\}$ in all possible ways by the rankings in $T$. Let $i_b$ denote the position of alternative $b \in A \setminus \{a\}$. Then,

$$L(V, a) = p_1^{\mathrm{SC}^{\mathrm{APP}}(a)} \cdot \sum_{\{i_b\}_{b \in A \setminus \{a\}} = \{2, \ldots, m\}} \prod_{b \in A \setminus \{a\}} p_{i_b}^{\mathrm{SC}^{\mathrm{APP}}(b)}$$

$$= (p_1)^{n \cdot k} \cdot \sum_{\{i_b\}_{b \in A \setminus \{a\}} = \{2, \ldots, m\}} \prod_{b \in A \setminus \{a\}} \left( \frac{p_{i_b}}{p_1} \right)^{\mathrm{SC}^{\mathrm{APP}}(b)}.$$

The second transition holds because $\text{SC}^{\text{APP}}(a) = n \cdot k - \sum_{b \in A \setminus \{a\}} \text{SC}^{\text{APP}}(b)$. Our assumption in the theorem statement implies $0 < p_{i_b}/p_1 < 1$ for $i_b \in \{2, \ldots, m\}$. Now, it can be checked that for $a, b \in A$, we have $\widehat{L}(V, a)/\widehat{L}(V, b) = \sum_{i \in \{2, \ldots, m\}} (p_i/p_1)^{\text{SC}^{\text{APP}}(b) - \text{SC}^{\text{APP}}(a)}$. Thus, $\text{SC}^{\text{APP}}(a) \geq \text{SC}^{\text{APP}}(b)$ if and only if $\widehat{L}(V, a) \geq \widehat{L}(V, b)$, as required. ∎

Note that the conditions of Theorem 2 are very mild. In particular, the condition for plurality votes is satisfied under the Mallows model with all five distances we consider, and the condition for veto votes is satisfied under the Mallows model with the Kendall tau, the footrule, and the maximum displacement distances. This is presented as Theorem 4 in the online appendix (Appendix B).

## 4  High Noise and Low Noise

While Theorem 1 shows that there are situations where at least some of the approval winners may not be MLE best alternatives, it does not paint the complete picture. In particular, in both profiles used as counterexamples in the proof of Theorem 1, it holds that every MLE best alternative is an approval winner. That is, the optimal rule choosing an MLE best alternative works as if a tie-breaking scheme is imposed on top of approval voting. Does this hold true for all profiles? Part 2 of Theorem 1 gives a positive answer for the Kendall tau distance. In this section, we answer the foregoing question (largely) in the positive under the other four distance functions, with respect to the two ends of the Mallows spectrum: the case of low noise ($\varphi \to 0$), and the case of high noise ($\varphi \to 1$). The case of high noise is especially compelling (because that is when it becomes hard to pinpoint the ground truth), but both extreme cases have received special attention in the literature [24, 21, 11]. In contrast to previous results, which have almost always yielded different answers in the two cases, we show that every MLE best alternative is an approval winner in *both cases*, in almost every situation.

We begin with the likelihood function for alternative $a$: $L(V, a) = \sum_{\sigma^*:\sigma^*(1)=a} \sum_{\pi:\pi \to V} \varphi^{d(\pi, \sigma^*)}$. When $\varphi \to 0$, maximizing $L(V, a)$ requires minimizing the minimum exponent. Ties, if any, are broken using the number of terms achieving the minimum exponent, then the second smallest exponent, and so on. At the other extreme, let $\varphi = 1 - \epsilon$ with $\epsilon \to 0$. Using the first-order approximation $(1 - \epsilon)^{d(\pi, \sigma^*)} \approx 1 - \epsilon \cdot d(\pi, \sigma^*)$, maximizing $L(V, a)$ requires minimizing the sum of $d(\pi, \sigma^*)$ over all $\sigma^*, \pi$ with $\sigma^*(1) = a$ and $\pi \to V$. Ties are broken using higher-order approximations. Let

$$L_0(V, a) = \min_{\sigma^*:\sigma^*(1)=a} \min_{\pi:\pi \to V} d(\pi, \sigma^*) \qquad L_1(V, a) = \sum_{\sigma^*:\sigma^*(1)=a} \sum_{\pi:\pi \to V} d(\pi, \sigma^*).$$

We are interested in minimizing $L_0(V, a)$ and $L_1(V, a)$; this leads to novel combinatorial problems that require detailed analysis. We are now ready for the main result of this section.

**Theorem 3.** *The following statements hold for using approval voting to aggregate $k$-approval votes drawn from the Mallows model.*

1. *Under the Mallows model with $d \in \{d_{FR}, d_{CY}, d_{HM}\}$ and $\varphi \to 0$, and under the Mallows model with $d \in \{d_{FR}, d_{CY}, d_{HM}, d_{MD}\}$ and $\varphi \to 1$, it holds that for every $k \in [m-1]$, and every profile with $k$-approval votes, every MLE best alternative is an approval winner.*

2. *Under the Mallows model with $d = d_{MD}$ and $\varphi \to 0$, there exists a profile with seven 2-approval votes over 5 alternatives such that no MLE best alternative is an approval winner.*

Before we proceed to the proof, we remark that in part 1 of the theorem, by $\varphi \to 0$ and $\varphi \to 1$, we mean that there exist $0 < \varphi_0^*, \varphi_1^* < 1$ such that the result holds for all $\varphi \leq \varphi_0^*$ and $\varphi \geq \varphi_1^*$, respectively. In part 2 of the theorem, we mean that for every $\varphi^* > 0$, there exists a $\varphi < \varphi^*$ for which the negative result holds. Due to space constraints, we only present the proof for the Mallows model with $d = d_{FR}$ and $\varphi \to 0$; the full proof appears in the online appendix (Appendix C).

*Proof of Theorem 3 (only for $d = d_{FR}, \phi \to 0$).* Let $\varphi \to 0$ in the Mallows model with the footrule distance. To analyze $L_0(V, \cdot)$, we first analyze $\min_{\pi:\pi \to V} d_{FR}(\sigma^*, \pi)$ for a fixed $\sigma^* \in \mathcal{L}(A)$. Then, we minimize it over $\sigma^*$, and show that the set of alternatives that appear first in the minimizers (i.e., the set of alternatives minimizing $L_0(V, a)$) is exactly the set of approval winners. Since every MLE best alternative in the $\varphi \to 0$ case must minimize $L_0(V, \cdot)$, the result follows.

Fix $\sigma^* \in \mathcal{L}(A)$. Imagine a boundary between positions $k$ and $k+1$ in all rankings, i.e., between the approved and the non-approved alternatives. Now, given a profile $\pi$ such that $\pi \rightarrow V$, we first apply the following operation repeatedly. For $i \in [n]$, let an alternative $a \in A$ be in positions $t$ and $t'$ in $\sigma^*$ and $\pi_i$, respectively. If $t$ and $t'$ are on the same side of the boundary (i.e., either both are at most $k$ or both are greater than $k$) and $t \neq t'$, then swap alternatives $\pi_i(t)$ and $\pi_i(t') = a$ in $\pi_i$. Note that this decreases the displacement of $a$ in $\pi_i$ with respect to $\sigma^*$ by $|t - t'|$, and increases the displacement of $\pi_i(t)$ by *at most* $|t - t'|$. Hence, the operation *cannot increase* $d_{FR}(\pi, \sigma^*)$. Let $\pi^*$ denote the profile that we converge to. Note that $\pi^*$ satisfies $\pi^* \rightarrow V$ (because we only swap alternatives on the same side of the boundary), $d_{FR}(\pi^*, \sigma^*) \leq d_{FR}(\pi, \sigma^*)$, and the following condition:

*Condition X*: for $i \in [n]$, every alternative that is on the same side of the boundary in $\sigma^*$ and $\pi_i^*$ is in the same position in both rankings.

Because we started from an arbitrary profile $\pi$ (subject to $\pi \rightarrow V$), it follows that it is sufficient to minimize $d_{FR}(\pi^*, \sigma^*)$ over all $\pi^*$ with $\pi^* \rightarrow V$ satisfying condition $X$. However, we show that subject to $\pi^* \rightarrow V$ and condition $X$, $d_{FR}(\pi^*, \sigma^*)$ is actually a constant.

Note that for $i \in [n]$, every alternative that is in different positions in $\pi_i^*$ and $\sigma^*$ must be on different sides of the boundary in the two rankings. It is easy to see that in every $\pi_i^*$, there is an equal number of alternatives on both sides of the boundary that are not in the same position as they are in $\sigma^*$. Now, we can divide the total footrule distance $d_{FR}(\pi^*, \sigma^*)$ into four parts:

1. Let $i \in [n]$ and $t \in [k]$ such that $\sigma^*(t) \neq \pi_i^*(t)$. Let $a = \sigma^*(t)$ and $(\pi_i^*)^{-1}(a) = t' > k$. Then, the displacement $t' - t$ of $a$ is broken into two parts: (i) $t' - k$, and (ii) $k - t$.

2. Let $i \in [n]$ and $t \in [m] \setminus [k]$ such that $\sigma^*(t) \neq \pi_i^*(t)$. Let $a = \sigma^*(t)$ and $(\pi_i^*)^{-1}(a) = t' \leq k$. Then, the displacement $t - t'$ of $a$ is broken into two parts: (i) $k - t'$, and (ii) $t - k$.

Because the number of alternatives of type $1$ and $2$ is equal for every $\pi_i^*$, we can see that the total displacements of types $1(i)$ and $2(ii)$ are equal, and so are the total displacements of types $1(ii)$ and $2(i)$. By observing that there are exactly $n - \mathrm{SC}^{\mathrm{APP}}(\sigma^*(t))$ instances of type $1$ for a given value of $t \leq k$, and $\mathrm{SC}^{\mathrm{APP}}(\sigma^*(t))$ instances of type $2$ for a given value of $t > k$, we conclude that

$$d_{FR}(\pi^*, \sigma^*) = 2 \cdot \left[ \sum_{t=1}^{k} (n - \mathrm{SC}^{\mathrm{APP}}(\sigma^*(t))) \cdot (k - t) + \sum_{t=k+1}^{m} \mathrm{SC}^{\mathrm{APP}}(\sigma^*(t)) \cdot (t - k) \right].$$

Minimizing this over $\sigma^*$ reduces to minimizing $\sum_{t=1}^{m} \mathrm{SC}^{\mathrm{APP}}(\sigma^*(t)) \cdot (t - k)$. By the rearrangement inequality, this is minimized when alternatives are ordered in a non-increasing order of their approval scores. Note that exactly the set of approval winners appear first in such rankings. ∎

Theorem 3 shows that under the Mallows model with $d \in \{d_{FR}, d_{CY}, d_{HM}\}$, every MLE best alternative is an approval winner for both $\varphi \rightarrow 0$ and $\varphi \rightarrow 1$. We believe that the same statement holds for all values of $\varphi$, as we were unable to find a counterexample despite extensive simulations.

**Conjecture 1.** *Under the Mallows model with distance $d \in \{d_{FR}, d_{CY}, d_{HM}\}$, every MLE best alternative is an approval winner for every $\varphi \in (0, 1)$.*

## 5 Experiments

We perform experiments with two real-world datasets — Dots and Puzzle [20] — to compare the performance of approval voting against that of the rule that is MLE *for the empirically observed distribution of $k$-approval votes* (and not for the Mallows model). Mao et al. [20] collected these datasets by asking workers on Amazon Mechanical Turk to rank either four images by the number of dots they contain (Dots), or four states of an 8-puzzle by their distance to the goal state (Puzzle). Hence, these datasets contain ranked votes over 4 alternatives in a setting where a true ranking of the alternatives indeed exists. Each dataset has four different noise levels; higher noise was created by increasing the task difficulty [20]. For Dots, ranking images with a smaller difference in the number of dots leads to high noise, and for Puzzle, ranking states farther away from the goal state leads to high noise. Each noise level of each dataset contains 40 profiles with approximately 20 votes each.

In our experiments, we extract 2-approval votes from the ranked votes by taking the top 2 alternatives in each vote. Given these 2-approval votes, approval voting returns an alternative with the largest number of approvals. To apply the MLE rule, however, we need to learn the underlying distribution of 2-approval votes. To that end, we partition the set of profiles in each noise level of each dataset into training (90%) and test (10%) sets. We use a high fraction of the profiles for training in order to examine the maximum advantage that the MLE rule may have over approval voting.

Given the training profiles (which approval voting simply ignores), the MLE rule learns the probabilities of observing each of the 6 possible 2-subsets of the alternatives given a fixed true ranking.[4] On the test data, the MLE rule first computes the likelihood of each ranking given the votes. Then, it computes the likelihood of each alternative being the best by adding the likelihoods of all rankings that put the alternative first. It finally returns an alternative with the highest likelihood.

We measure the accuracy of both methods by their frequency of being able to pinpoint the correct best alternative. For each noise level in each dataset, the accuracy is averaged over 1000 simulations with random partitioning of the profiles into training and test sets.

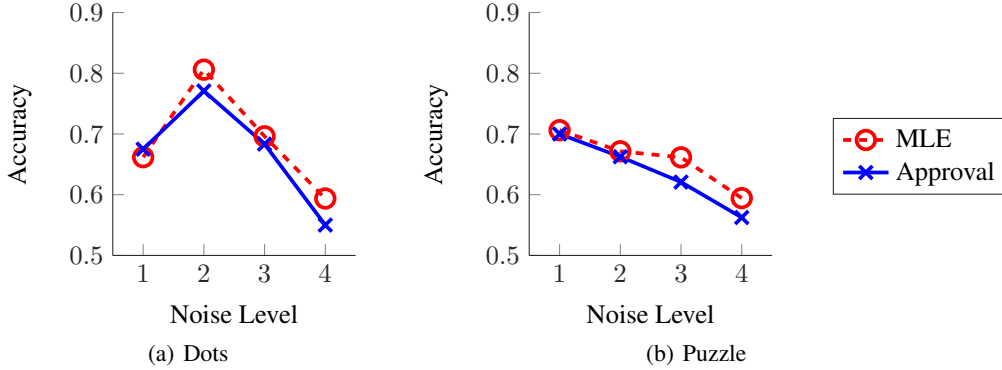

(a) Dots

(b) Puzzle

Fig. 1: The MLE rule (trained on 90% of the profiles) and approval voting for 2-approval votes.

Figures 1(a) and 1(b) show that in general the MLE rule does achieve greater accuracy than approval voting. However, the increase is at most 4.5%, which may not be significant in some contexts.

## 6    Discussion

Our main conclusion from the theoretical and empirical results is that approval voting is typically close to optimal for aggregating $k$-approval votes. However, the situation is much subtler than it appears at first glance. Moreover, our theoretical analysis is restricted by the assumption that the votes are drawn from the Mallows model. A recent line of work in social choice theory [9, 10] has focused on designing voting rules that perform well — simultaneously — under a wide variety of noise models. It seems intuitive that approval voting would work well for aggregating $k$-approval votes under any reasonable noise model; an analysis extending to a wide family of realistic noise models would provide a stronger theoretical justification for using approval voting.

On the practical front, it should be emphasized that approval voting is *not always optimal*. When maximum accuracy matters, one may wish to switch to the MLE rule. However, learning and applying the MLE rule is much more demanding. In our experiments we learn the entire distribution over $k$-approval votes given the true ranking. While for 2-approval or 3-approval votes over 4 alternatives we only need to learn 6 probability values, in general for $k$-approval votes over $m$ alternatives one would need to learn $\binom{m}{k}$ probability values, and the training data may not be sufficient for this purpose. This calls for the design of estimators for the best alternative that achieve greater statistical efficiency by avoiding the need to learn the entire underlying distribution over votes.

## Footnotes

[1] $k$-approval is also used for picking $k$ winners, e.g., various cities in the US such as San Francisco, Chicago, and New York use it in their so-called "participatory budgeting" process [15].

[2] There is a subtle distinction, which we will not belabor, between $k$-approval voting, which is the focus of this paper, and approval voting [8], which allows voters to approve as many alternatives as they wish. The latter

[3]Technically, this is a social choice function; a social welfare function returns a ranking of the alternatives.

[4]Technically, we learn a neutral noise model where the probability of a subset of alternatives being observed only depends on the positions of the alternatives in the true ranking.

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
