[Supplementary Material · appendix.pdf]

## Online Appendix To: Is Approval Voting Optimal Given Approval Votes?

## A   Calculations for the Counterexamples in Theorem 1

Recall from Equation (1) that the likelihood of an alternative $a$ given $n$ ($k$-approval) votes $V$ is proportional to

$$L(V, a) \triangleq \sum_{\sigma^*:\sigma^*(1)=a} \sum_{\pi:\pi \to V} \varphi^{d(\pi,\sigma^*)} = \sum_{\sigma^*:\sigma^*(1)=a} \prod_{i=1}^{n} \left[ \sum_{\pi_i:\pi_i \to V_i} \varphi^{d(\pi_i,\sigma^*)} \right].$$

In this section, we illustrate how to use this formula to verify the counterexamples used in the proof of Theorem 1. We show the detailed procedure in the case of the counterexample used for distances $d_{MD}, d_{CY}$, and $d_{HM}$. The counterexample for $d_{FR}$ can be verified similarly.

Recall the counterexample for $d \in \{d_{MD}, d_{CY}, d_{HM}\}$. We have 5 voters providing 2-approval votes over the set of 4 alternatives $A = \{a, b, c, d\}$, where

$$V_1 = V_2 = \{b, c\}, \; V_3 = \{a, d\}, \; V_4 = \{a, b\}, \; V_5 = \{a, c\}.$$

We want to verify that under the Mallows model with $d \in \{d_{MD}, d_{CY}, d_{HM}\}$, an approval winner (in particular, $a$) is not an MLE best alternative for any value of $\varphi \in (0, 1)$ by showing that a different alternative $b$ has greater likelihood. Let $T_a^*$ (resp. $T_b^*$) denote the set of rankings of $A$ that place $a$ (resp. $b$) in the first position, i.e.,

$$\begin{aligned}
T_a^* = \{ & a \succ b \succ c \succ d, a \succ b \succ d \succ d, \\
& a \succ c \succ b \succ d, a \succ c \succ d \succ b, \\
& a \succ d \succ b \succ c, a \succ d \succ c \succ b\}, \\
T_b^* = \{ & b \succ a \succ c \succ d, b \succ a \succ d \succ d, \\
& b \succ c \succ a \succ d, b \succ c \succ d \succ a, \\
& b \succ d \succ a \succ c, b \succ d \succ c \succ a\}.
\end{aligned}$$

For $i \in [5]$, let $T_i = \{\pi_i \in \mathcal{L}(A) | \pi_i \to V_i\}$ denote the set of rankings that could generate the $k$-approval vote $V_i$, i.e.,

$$\begin{aligned}
T_1 = T_2 &= \{b \succ c \succ a \succ d, b \succ c \succ d \succ a, c \succ b \succ a \succ d, c \succ b \succ d \succ a\}, \\
T_3 &= \{a \succ d \succ b \succ c, a \succ d \succ c \succ b, d \succ a \succ b \succ c, d \succ a \succ c \succ b\}, \\
T_4 &= \{a \succ b \succ d \succ c, a \succ b \succ c \succ d, b \succ a \succ d \succ c, b \succ a \succ c \succ d\}, \\
T_5 &= \{a \succ c \succ d \succ b, a \succ c \succ b \succ d, c \succ a \succ d \succ b, c \succ a \succ b \succ d\}.
\end{aligned}$$

Using these, we can now evaluate the likelihoods

$$L(V, a) = \sum_{\sigma^* \in T_a^*} \prod_{i=1}^{5} \left[ \sum_{\pi_i \in T_i} \varphi^{d(\pi_i,\sigma^*)} \right], \quad L(V, b) = \sum_{\sigma^* \in T_b^*} \prod_{i=1}^{5} \left[ \sum_{\pi_i \in T_i} \varphi^{d(\pi_i,\sigma^*)} \right].$$

Using a computer program, it can be verified that for $d \in \{d_{MD}, d_{HM}, d_{CY}\}$, the difference in likelihoods $L(V, b) - L(V, a)$, which is a polynomial in $\varphi$, has no roots in $(0, 1)$, and is positive at $\varphi = 0.5$. Hence, due to the intermediate value theorem, it must be positive in the entire interval. Hence, $a$ is not an MLE best alternative for any value of $\varphi \in (0, 1)$, as required.

## B   Mildness of the Conditions in Theorem 2

In this section, we want to prove that the conditions in Theorem 2 are mild, by showing that they hold for popular distance metrics. Let $T_{i,j}(\sigma^*) = \{\sigma \in \mathcal{L}(A) | \sigma^{-1}(\sigma^*(i)) \leq j\}$ be the set of rankings where alternative in position $i$ in $\sigma^*$ appears among the top $j$ positions. Caragiannis et al. [9] study three of our distance metrics: the Kendall tau distance, the footrule distance, and the maximum displacement distance. Combining their Lemma 5.5 and Theorem 5.9, one gets the following lemma.

**Lemma 1** ([9]). *For $d \in \{d_{KT}, d_{FR}, d_{MD}\}$, $\sigma^* \in \mathcal{L}(A)$, $i, i' \in [m]$ with $i < i'$, and $j \in [m-1]$, there exists a bijection $f : T_{i,j}(\sigma^*) \to T_{i',j}(\sigma^*)$ such that $d(f(\sigma), \sigma^*) \geq d(\sigma, \sigma^*)$ for all $\sigma \in T_{i,j}(\sigma^*)$, and $d(f(\sigma), \sigma^*) > d(\sigma, \sigma^*)$ for at least one $\sigma \in T_{i,j}(\sigma^*)$.*

That is, rankings placing a less preferred alternative among the top positions are generally farther away from $\sigma^*$; this is highly intuitive. We use this property to establish our required conditions for the three distance metrics $\{d_{KT}, d_{FR}, d_{MD}\}$. For the Cayley and the Hamming distances, unfortunately, Caragiannis et al. [9] give an example demonstrating that they violate the property in Lemma 1. For these metrics, we give a different result that is sufficient for our purpose. First, we need the following characterization (see, e.g., [9]) of the Cayley distance.

**The Cayley distance:** Given two rankings $\sigma_1$ and $\sigma_2$, suppose we want to convert $\sigma_1$ to $\sigma_2$ by pairwise swaps of alternatives. We create a directed graph (let us call it the conversion graph) over the alternatives where each alternative $a$ points to the alternative $\sigma_1(\sigma_2^{-1}(a))$, that is, the alternative whose position in $\sigma_1$ matches the position of $a$ in $\sigma_2$. This indicates that we need to move $a$ to the position of this alternative in $\sigma_1$. It can be checked that this graph is a collection of disjoint directed cycles, and the Cayley distance between the two rankings is $m$ minus the number of cycles.

We are now ready to prove two useful results.

**Lemma 2.** *Let $d \in \{d_{CY}, d_{HM}\}$, $\sigma_1, \sigma_2 \in \mathcal{L}(A)$, and $i, j \in [m]$. For $t \in [2]$, let $\widehat{\sigma}_t = (\sigma_t)_{\sigma_t(i) \leftrightarrow \sigma_t(j)}$ denote the ranking obtained by swapping alternatives at positions $i$ and $j$ in $\sigma_t$. Then, $d(\sigma_1, \sigma_2) = d(\widehat{\sigma_1}, \widehat{\sigma_2})$.*

*Proof.* For the Hamming distance, this follows immediately from the definition. For the Cayley distance, the result follows directly from the definition of the conversion graph. Indeed, when we swap alternatives in positions $i$ and $j$ in two rankings $\sigma_1$ and $\sigma_2$, for every alternative $a$ the alternative in position $\sigma_2^{-1}(a)$ in $\sigma_1$ is the same, before or after the swap. In other words, the function $\sigma_1(\sigma_2^{-1}(\cdot))$, which induces the conversion graph, remains intact. Hence, the Cayley distance remains intact, as required. ∎ (Proof of Lemma 2)

Note that swapping alternatives in two fixed positions in two rankings is different from swapping two fixed alternatives. Hence, Lemma 2 is not satisfied by all neutral distance metrics.

**Lemma 3.** *For $d \in \{d_{CY}, d_{HM}\}$, $\sigma \in \mathcal{L}(A)$, and $i \in [m-1]$, there exists a bijection $f : T_{i,1}(\sigma^*) \to T_{i+1,1}(\sigma^*)$ such that*

1. *for $i \geq 2$, $d(f(\sigma), \sigma^*) = d(\sigma, \sigma^*)$ for all $\sigma \in T_{i,1}(\sigma^*)$, and*

2. *for $i = 1$, $d(f(\sigma), \sigma^*) > d(\sigma, \sigma^*)$ for all $\sigma \in T_{1,1}(\sigma^*)$.*

*Proof.* **The Hamming distance, i $\geq$ 2:** Let $d = d_{HM}$. Fix rankings $\sigma^* \in \mathcal{L}(A)$ and $\sigma \in T_{i,1}(\sigma^*)$. Let $a_i = \sigma^*(i) = \sigma(1)$ and $a_{i+1} = \sigma^*(i+1)$. Next, let $\widehat{\sigma}$ and $\widehat{\sigma}^*$ denote the rankings obtained by swapping the alternatives in positions $i$ and $i+1$ in $\sigma$ and $\sigma^*$, respectively. Lemma 2 implies that $d(\sigma, \sigma^*) = d(\widehat{\sigma}, \widehat{\sigma}^*)$. Further, $i \geq 2$ implies that $\widehat{\sigma}(1) = \sigma(1) = a_i$. Also, $\widehat{\sigma}^*(i+1) = a_i$ and $\widehat{\sigma}^*(i) = a_{i+1}$. Finally, we exchange the labels of alternatives $a_i$ and $a_{i+1}$. Applying this operation on $\widehat{\sigma}^*$ yields $\sigma^*$ back. Let $\tilde{\sigma}$ denote the ranking obtained by applying this operation to $\widehat{\sigma}$. Then, clearly, $\tilde{\sigma}(1) = a_{i+1} = \sigma^*(i+1)$. Hence, $\tilde{\sigma} \in T_{i+1,1}(\sigma^*)$. Due to the neutrality of the Hamming distance, $d(\tilde{\sigma}, \sigma^*) = d(\widehat{\sigma}, \widehat{\sigma}^*) = d(\sigma, \sigma^*)$. The proof is complete by assigning $f(\sigma) = \tilde{\sigma}$ for every $\sigma \in T_{i,1}(\sigma^*)$. The fact that $f$ is a bijection follows from the observation that its two parts — swapping alternatives in columns $i$ and $i+1$, and exchanging the labels of alternatives $a_i$ and $a_{i+1}$ — are bijections themselves.

**The Cayley distance, i $\geq$ 2:** Let $d = d_{CY}$. For the Cayley distance, we use the same bijection that we used for the Hamming distance. Once again, the first operation (swapping alternatives in positions $i$ and $i+1$) keeps the Cayley distance intact due to Lemma 2, and the second operation (exchanging labels of alternatives $a_i$ and $a_{i+1}$) keeps the Cayley distance intact due to the neutrality of the metric. Hence, we get that $d(f(\sigma), \sigma^*) = d(\sigma, \sigma^*)$, as required.

**The Hamming distance, i = 1:** Let $d = d_{HM}$. Let $\sigma^* \in \mathcal{L}(A)$ and $\sigma \in T_{1,1}(\sigma^*)$. Again, let $a_1 = \sigma^*(1) = \sigma(1)$ and $a_2 = \sigma^*(2)$. Let $t = \sigma^{-1}(a_2)$. Let $f(\sigma) = \sigma_{a_1 \leftrightarrow a_2}$. Observe that not only

$f(\sigma) \in T_{2,1}(\sigma^*)$, but $f$ is a bijection from $T_{1,1}(\sigma^*)$ to $T_{2,1}(\sigma^*)$. Next, note that only positions 1 and $t$ are affected when we swap alternatives $a_1$ and $a_2$ in $\sigma$. Hence, to show $d(f(\sigma), \sigma^*) > d(\sigma, \sigma^*)$, we only need to establish that

$$\mathbb{I}[f(\sigma)(1) \neq \sigma^*(1)] + \mathbb{I}[f(\sigma)(t) \neq \sigma^*(t)] > \mathbb{I}[\sigma(1) \neq \sigma^*(1)] + \mathbb{I}[\sigma(t) \neq \sigma^*(t)]. \qquad (4)$$

However, in the LHS, we have $\mathbb{I}[f(\sigma)(1) \neq \sigma^*(1)] = \mathbb{I}[a_2 \neq a_1] = 1$ and $\mathbb{I}[f(\sigma)(t) \neq \sigma^*(t)] = \mathbb{I}[a_1 \neq \sigma^*(t)] = 1$, whereas in the RHS, we have $\mathbb{I}[\sigma(1) \neq \sigma^*(1)] = \mathbb{I}[a_1 \neq a_1] = 0$. Hence, Equation (4) holds, as required.

**The Cayley distance, i = 1:** Let $d = d_{CY}$. Let $\sigma^* \in \mathcal{L}(A)$ and $\sigma \in T_{1,1}(\sigma^*)$. Again, let $a_1 = \sigma^*(1) = \sigma(1)$ and $a_2 = \sigma^*(2)$. Define $f(\sigma) = \sigma_{a_1 \leftrightarrow a_2}$. Once again, $f$ is a bijection from $T_{1,1}(\sigma^*)$ to $T_{2,1}(\sigma^*)$. Next, let us compare two conversion graphs: graph $G_1$ for converting $\sigma$ to $\sigma^*$, and graph $G_2$ for converting $f(\sigma)$ to $\sigma^*$. It can be checked that $G_2$ is identical to $G_1$ except that, instead of having a self-loop at $a_1$ as in $G_1$, $a_1$ has an incoming edge that was originally incoming to $a_2$ in $G_1$, and $a_1$ has an outgoing edge to $a_2$. In other words, $G_1$ has a self-loop at $a_1$, which, in $G_2$, is absorbed into the loop that contains $a_2$. Thus, $G_2$ has one less cycle than $G_1$, i.e., $d(f(\sigma), \sigma^*) > d(\sigma, \sigma^*)$, as required. Note that this is also true in the special case where $\sigma(2) = a_2$. In that case, $G_1$ has self-loops at both $a_1$ and $a_2$, whereas $G_2$ has a 2-cycle between $a_1$ and $a_2$. ∎ (Proof of Lemma 3)

We are now ready to analyze the conditions for plurality and veto votes in Theorem 2.

**Theorem 4.** *For $i \in [m]$, let $p_i$ and $q_i$ denote the probabilities of the alternative in position $i$ in the true ranking $\sigma^*$ appearing in the first and the last positions, respectively, under the Mallows model with distance metric $d$. Then,*

1. *For $d \in \{d_{KT}, d_{FR}, d_{MD}\}$, we have $p_1 > p_i > 0$ and $0 < q_1 < q_i$ for $i \in \{2, \ldots, m\}$.*

2. *For $d \in \{d_{CY}, d_{HM}\}$, we have $p_1 > p_i > 0$ for $i \in \{2, \ldots, m\}$.*

*Proof.* Note that $p_i, q_i > 0$ holds trivially for all $i \in [m]$ because under the Mallows model (with any distance metric and $\varphi > 0$), every ranking has a positive probability. For $d \in \{d_{KT}, d_{FR}, d_{MD}\}$, we directly leverage Lemma 1. Fix $\sigma^* \in \mathcal{L}(A)$. Let $f$ denote the bijection established in Lemma 1 from $T_{i,1}(\sigma^*)$ to $T_{i+1,1}(\sigma^*)$. For $i \in [m-1]$, under the Mallows model we have

$$
\begin{aligned}
p_i - p_{i+1} &= \frac{1}{Z_\varphi^m} \cdot \left[ \sum_{\sigma \in T_{i,1}(\sigma^*)} \varphi^{d(\sigma, \sigma^*)} - \sum_{\sigma \in T_{i+1,1}(\sigma^*)} \varphi^{d(\sigma, \sigma^*)} \right] \\
&= \frac{1}{Z_\varphi^m} \cdot \sum_{\sigma \in T_{i,1}(\sigma^*)} \left[ \varphi^{d(\sigma, \sigma^*)} - \varphi^{d(f(\sigma), \sigma^*)} \right] \\
&> 0,
\end{aligned}
$$

where the last inequality follows because Lemma 1 ensures $d(f(\sigma, \sigma^*)) \geq d(\sigma, \sigma^*)$ for every $\sigma \in T_{i,1}(\sigma^*)$, and $d(f(\sigma, \sigma^*)) > d(\sigma, \sigma^*)$ for at least one $\sigma \in T_{i,1}(\sigma^*)$. Hence, $p_i > p_{i+1}$ for all $i \in [m-1]$, which directly implies $p_1 > \max_{i \in \{2, \ldots, m\}} p_i$.

Using a similar argument, we can also see that for $i \in [m-1]$,

$$(1 - q_i) - (1 - q_{i+1}) = \frac{1}{Z_\varphi^m} \cdot \left[ \sum_{\sigma \in T_{i,m-1}(\sigma^*)} \varphi^{d(\sigma, \sigma^*)} - \sum_{\sigma \in T_{i+1,m-1}(\sigma^*)} \varphi^{d(\sigma, \sigma^*)} \right] > 0.$$

Hence, we have that $q_i < q_{i+1}$ for all $i \in [m-1]$; it follows that $q_1 < \min_{i \in \{2, \ldots, m\}} q_i$, as required.

The argument for the Hamming and the Cayley distances, i.e., for $d \in \{d_{HM}, d_{CY}\}$, is similar, but we use Lemma 3 instead of Lemma 1. It can be checked that this leads to $p_1 > p_2$ and $p_i = p_{i+1}$ for $i \in \{2, \ldots, m-1\}$. This still implies $p_1 > \max_{i \in \{2, \ldots, m\}} p_i$, as required. ∎ (Proof of Theorem 4)

Caragiannis et al. [9] show (in Appendix D.4 in the full version of their paper) that in the case of the Cayley distance (resp., the Hamming distance) with $m = 3$, both $\sigma^*(1)$ and $\sigma^*(2)$ appear last

in exactly two rankings: one at distance 1 (resp., 3) from $\sigma^*$, and another at distance 2 from $\sigma^*$. Hence, under any noise model where the probability of a ranking is a function of its distance from $\sigma^*$ (e.g., under the Mallows model), we have $q_1 = q_2$, which violates the condition for veto votes in Theorem 2.

## C   Continued Proof of Theorem 3

While for the footrule distance we could show that only approval winners minimize $L_0(V, a)$, for the Cayley and the Hamming distances it can be shown that even alternatives with suboptimal approval score might sometimes minimize $L_0(V, a)$. In many cases, several higher orders of tie-breaking do not help distinguish approval winners from other alternatives. Hence, we devise a way to completely avoid the analysis of multiple levels of tie-breaking.

First, we say that a set $S_1$ is lexicographically smaller than another set $S_2$ if, after sorting their elements in non-decreasing order and comparing from the lowest to the highest, when they differ for the first time $S_1$ has a smaller element than $S_2$. Recall that in the $\varphi \to 0$ case, we can determine the MLE best alternative as follows. We consider the set of distances $D_a = \{d(\pi, \sigma^*)\}_{\sigma^*:\sigma^*(1)=a, \pi:\pi \to V}$ for each alternative $a$. Then, the alternative whose set is lexicographically smallest is the MLE best alternative. There may be multiple tied MLE best alternatives whose sets are identical. We are now ready to prove the following simple and useful lemma.

**Lemma 4.** *Let distance metric $d \in \{d_{CY}, d_{HM}\}$. For alternatives $a, b \in A$, define*

$$m_{a,-b} = \min_{\substack{\sigma^*:\sigma^*(1)=a \wedge (\sigma^*)^{-1}(b)>k \\ \pi:\pi \to V}} d(\pi, \sigma^*).$$

*Then, there exists $\varphi^* > 0$ such that for all $0 < \varphi \le \varphi^*$, alternative $a$ has higher likelihood than alternative $b$ if $m_{a,-b} < m_{b,-a}$.*

*Proof.* Let $\mathcal{L}_{a,b}(A) = \{\sigma^* \in \mathcal{L}(A) | \sigma^*(1) = a \wedge (\sigma^*)^{-1}(b) \le k$ (i.e., the set of rankings that put $a$ first and $b$ among the top $k$ positions), and $\pi = \{\pi | \pi \to V\}$. For $\sigma^* \in \mathcal{L}_{a,b}(A)$ and $\pi \in \pi$, let $\widehat{\sigma}^*$ and $\widehat{\pi}$ denote the ranking and the profile obtained by swapping alternatives at positions 1 and $(\sigma^*)^{-1}(b)$ in $\sigma^*$ and in every $\pi_i \in \pi$, respectively. Then, it can be checked that $(\sigma^*, \pi) \leftrightarrow (\widehat{\sigma}^*, \widehat{\pi})$ is a bijection from $\mathcal{L}_{a,b}(A) \times \pi$ to $\mathcal{L}_{b,a}(A) \times \pi$. Further, Lemma 2 implies that $d(\pi, \sigma^*) = d(\widehat{\pi}, \widehat{\sigma}^*)$. Hence, when comparing $D_a$ to $D_b$, rankings that put $a$ first and $b$ among the top $k$ positions cancel out with rankings that put $b$ first and $a$ among the top $k$ positions. The result now follows immediately. ∎ (Proof of Lemma 4)

We now use Lemma 4 to analyze the Cayley and the Hamming distances.

$\varphi \to 0$, **the Cayley and the Hamming distances:** We show that if alternative $a$ is an approval winner while alternative $b$ is not, then there exists $\varphi^* > 0$ such that for all $0 < \varphi \le \varphi^*$, alternative $a$ has a higher likelihood of being the best alternative than $b$. Due to Lemma 4, it is sufficient to show that $m_{a,-b} < m_{b,-a}$.

Fix a ranking $\sigma^* \in \mathcal{L}(A)$. We can show that the profile $\pi$ with $\pi \to V$ that minimizes the distance from $\sigma^*$ must also satisfy condition $X$ in the proof for the footrule distance. For $i \in [n]$, define $A_i = (V_i \cap \sigma^*([k])) \cup ((A \setminus V_i) \cap \sigma^*([m] \setminus [k]))$. In words, $A_i$ is the set of alternatives that we can place in the same position in $\pi_i$ as in $\sigma^*$ given the restriction $\pi_i \to V_i$. Condition $X$ says that $\pi_i$ and $\sigma^*$ would indeed agree on the positions of the alternatives in $A_i$, for each $i \in [n]$.

For the Hamming distance, observe that the restriction $\pi \to V$ ensures that $d_{HM}(\pi_i, \sigma^*) \ge |A \setminus A_i|$. The lower bound is achieved by satisfying condition $X$, i.e., putting all alternatives in $A_i$ in the same positions in $\pi_i$ as they are in $\sigma^*$.

For the Cayley distance, note that

$$d_{CY}(\pi_i, \sigma^*) \ge \frac{1}{2} \cdot d_{HM}(\pi_i, \sigma^*) \ge \frac{1}{2} \cdot |A \setminus A_i|.$$

Once again, to achieve this lower bound, we must align $\pi_i$ with $\sigma^*$ on the alternatives in $A_i$, and pair up alternatives from $A \setminus A_i$ in $\pi_i$ such that for every $a \in A \setminus A_i$, there exists a unique $b \in A \setminus A_i$

with $\pi_i((\sigma^*)^{-1}(a)) = b$ and $\pi_i((\sigma^*)^{-1}(b)) = a$. The pairing is possible because the number of alternatives from $A \setminus A_i$ on each side of the boundary in $\pi_i$ must be equal. Therefore, the lower bound on the distance can indeed be achieved by swapping these paired alternatives.

Define

$$P(\sigma^*) = \sum_{i=1}^{n} |A \setminus A_i| = \sum_{t=1}^{k} (n - \mathrm{SC}^{\mathrm{APP}}(\sigma^*(t))) + \sum_{t=k+1}^{m} \mathrm{SC}^{\mathrm{APP}}(\sigma^*(t)).$$

Then, we have established that for the Hamming distance, $\min_{\pi:\pi \to V} d_{HM}(\sigma^*, \pi) = P(\sigma^*)$, and for the Cayley distance, $\min_{\pi:\pi \to V} d_{CY}(\sigma^*, \pi) = (1/2) \cdot P(\sigma^*)$. Further, note that $P(\sigma^*) = C - \sum_{t=1}^{k} \mathrm{SC}^{\mathrm{APP}}(\sigma^*(t))$, where $C$ is independent of $\sigma^*$. Hence, minimizing $\min_{\pi:\pi \to V} d(\sigma^*, \pi)$ reduces to, in both cases, maximizing $\sum_{t=1}^{k} \mathrm{SC}^{\mathrm{APP}}(\sigma^*(t))$.

We are not done yet, because a ranking can maximize the sum of approval scores of its top $k$ alternatives while not placing the alternative with the maximum approval score at the top. However, Lemma 4 comes to our rescue. Let

$$w_{a,-b} = \max_{\sigma^*:\sigma^*(1)=a \wedge (\sigma^*)^{-1}(b)>k} \sum_{t=1}^{k} \mathrm{SC}^{\mathrm{APP}}(\sigma^*(t)).$$

Then, we only need to show that $w_{a,-b} > w_{b,-a}$ for any two alternatives $a, b \in A$ such that $a$ is an approval winner, while $b$ is not.

By the choice of $a$ and $b$, $\mathrm{SC}^{\mathrm{APP}}(a) > \mathrm{SC}^{\mathrm{APP}}(b)$. For $i \in [m]$, let $S_i$ denote the sum of the $i$ highest approval scores. If the approval score of $b$ is among the $k$ highest approval scores, then clearly $w_{a,-b} = S_{k+1} - \mathrm{SC}^{\mathrm{APP}}(b)$ and $w_{b,-a} = S_{k+1} - \mathrm{SC}^{\mathrm{APP}}(a)$. Hence, $w_{a,-b} > w_{b,-a}$, as required.

If the approval score of $b$ is not among the $k$ highest approval scores, then there are two cases.

1. *There are at most $k$ approval winners.* In this case, we can see that $w_{a,-b} = S_k$ while $w_{b,-a} < S_k$ (because in the latter case, we need to find a ranking that does not put $a$ among the top $k$ positions).

2. *There are $t$ approval winners, where $t > k$.* In this case, while $w_{a,-b} = w_{b,-a} = S_k$, and thus the exponents $m_{a,-b}$ and $m_{b,-a}$ in the likelihood expression — Equation (1) — are equal, the number of terms achieving this exponent in the likelihood function of $b$ is proportional to $\binom{t}{k}$, while the number of terms achieving this exponent in the likelihood function of $a$ is proportional to $\binom{t-1}{k}$ (because in the latter case, an approval winner — $a$ — is not allowed to be among the top $k$ positions). Hence, the tie is broken in favor of $a$.

### $\varphi \to 0$, the maximum displacement distance:

Consider the following vote profile $V$ over the set of alternatives $A = \{a, b, c, d, e\}$.

$$V_1 = \{a, b\} \quad V_2 = \{a, c\} \quad V_3 = \{b, c\} \quad V_4 = \{c, d\} \quad V_5 = \{c, e\} \quad V_6 = V_7 = \{d, e\}.$$

Note that alternative $c$ is the unique approval winner with an approval score of 4. However, it can be checked that $L_0(V, d) = L_0(V, e) = 11 < 13 = L_0(V, c)$. Hence, neither of the two MLE best alternatives ($d$ and $e$) is an approval winner, as required.

### $\varphi \to 1$, all five distances:
The case of $\varphi \to 1$ is easier because, as we will show, the analysis of $L_1(V, \cdot)$ is sufficient to differentiate approval winners from other alternatives for all five distance metrics. Let the distance metric be denoted $d \in \{d_{KT}, d_{FR}, d_{CY}, d_{HM}, d_{MD}\}$. Note that for an alternative $a \in A$,

$$L_1(V, a) = \sum_{\sigma^*:\sigma^*(1)=a} \sum_{\pi:\pi \to V} d(\pi, \sigma^*)$$

$$= \sum_{\sigma^*:\sigma^*(1)=a} \sum_{i=1}^{n} \sum_{\pi_i:\pi_i \to V_i} d(\pi_i, \sigma^*)$$

$$= \sum_{i=1}^{n} \sum_{\pi_i:\pi_i \to V_i} \left( \sum_{\sigma^*:\sigma^*(1)=a} d(\pi_i, \sigma^*) \right). \tag{5}$$

Now, define $S_t(\pi_i) = \sum_{\sigma^* \in \mathcal{L}(A):\sigma^*(1)=\pi_i(t)} d(\sigma^*, \pi_i)$. That is, $S_t(\pi_i)$ is the sum of distances of $\pi_i$ from all rankings that put its $t^{th}$ ranked alternative first. Due to neutrality of the distance metric $d$, this quantity is independent of the ranking $\pi_i$. Substituting this into Equation (5), we get

$$
\begin{aligned}
L_1(V, a) &= \sum_{i=1}^{n} \sum_{\pi_i:\pi_i \to V_i} S_{\pi_i^{-1}(a)} \\
&= \sum_{i=1}^{n} \mathbb{I}[a \in V_i] \cdot \left[ k! \cdot (m-k-1)! \cdot \sum_{j=1}^{k} S_j \right] + \mathbb{I}[a \notin V_i] \cdot \left[ (k-1)! \cdot (m-k)! \cdot \sum_{j=k+1}^{m} S_j \right] \\
&= k! \cdot (m-k)! \cdot \left[ \text{SC}^{\text{APP}}(a) \cdot \frac{\sum_{j=1}^{k} S_j}{k} + (n - \text{SC}^{\text{APP}}(a)) \cdot \frac{\sum_{j=k+1}^{m} S_j}{m-k} \right] \\
&= k! \cdot (m-k)! \cdot n \cdot \frac{\sum_{j=k+1}^{m} S_j}{m-k} + k! \cdot (m-k)! \cdot \text{SC}^{\text{APP}}(a) \cdot \left[ \frac{\sum_{j=1}^{k} S_j}{k} - \frac{\sum_{j=k+1}^{m} S_j}{m-k} \right].
\end{aligned}
$$

To show that this quantity is minimized when $a$ is an approval winner, we need to show that

$$
P_k = \frac{\sum_{j=1}^{k} S_j}{k} - \frac{\sum_{j=k+1}^{m} S_j}{m-k} \tag{6}
$$

satisfies $P_k < 0$ for all $k \in [m-1]$. To that end, we show the following lemma.

**Lemma 5.** *The following statements hold.*

1. *For distance $d \in \{d_{KT}, d_{FR}, d_{MD}\}$, we have $S_i < S_{i+1}$ for all $i \in [m-1]$.*

2. *For distance $d \in \{d_{CY}, d_{HM}\}$, we have $S_1 < S_2$ and $S_i = S_{i+1}$ for all $i \in \{2, \dots, m-1\}$.*

*Proof.* Consider a bijection $f$ from $T_{i,1}(\sigma^*)$ to $T_{i+1,,1}(\sigma^*)$. Then, for $i \in [m-1]$,

$$
\begin{aligned}
S_{i+1} - S_i &= \sum_{\sigma \in T_{i+1,1}(\sigma^*)} d(\sigma, \sigma^*) - \sum_{\sigma \in T_{i,1}(\sigma^*)} d(\sigma, \sigma^*) \\
&= \sum_{\sigma \in T_{i,1}(\sigma^*)} d(f(\sigma), \sigma^*) - d(\sigma, \sigma^*).
\end{aligned}
$$

Now, for $d \in \{d_{KT}, d_{FR}, d_{MD}\}$, Lemma 1 ensures the existence of a bijection $f$ for which $d(f(\sigma), \sigma^*) \geq d(\sigma, \sigma^*)$ for all $\sigma \in T_{i,1}(\sigma^*)$, and $d(f(\sigma), \sigma^*) > d(\sigma, \sigma^*)$ for at least one $\sigma \in T_{i,1}(\sigma^*)$. Hence, it follows that $S_{i+1} > S_i$ for all $i \in [m-1]$, as required.

For $d \in \{d_{CY}, d_{HM}\}$, Lemma 3 ensures the existence of a bijection $f$ such that for all $\sigma \in T_{i,1}(\sigma^*)$, we have $d(f(\sigma), \sigma^*) = d(\sigma, \sigma^*)$ if $i \geq 2$, and $d(f(\sigma), \sigma^*) > d(\sigma, \sigma^*)$ if $i = 1$. Hence, it follows that $S_2 > S_1$ and $S_{i+1} = S_i$ for all $i \in \{2, \dots, m-1\}$. ∎ (Proof of Lemma 5)

Using Lemma 5, it is easy to check that $P_k < 0$ in Equation (6), as required. ∎ (Proof of Theorem 3)