[Reviews · NeurIPS 2015]

Submitted by Assigned_Reviewer_1

Summary of the problem setting:

The paper considers a setting of crowdsourcing where every question is posed in an approval voting setting with the worker asked to select precisely k of the alternatives. The most popular way of aggregating these answers or votes is to simply select the alternative that received the maximum number of votes. It is generally unclear how to break ties, and often people adopt a procedure that gives an equal weight to all alternatives in the tie, for instance, by choosing one of them at random.

----------------- Summary of the contributions:

1) The authors prove that under the Mallows model with Kendall tau distance, the set of alternatives with the most votes coincides with the maximum likelihood estimate.

2) The authors give a counter example showing that for the Mallows model with four other distance functions, not all alternatives in the tie may be the maximum likelihood estimates. This leads to the assertion that there is additional structure in the problem which must be exploited to break ties.

----------------- Pros:

The problem is important and the results are fundamental given that both approval voting and the Mallows model are quite popular models. In fact, I was quite surprised that this very problem is not studied previously in the literature. Given the many negative results associated to the Mallows model and the Kemeny distance (in terms of computational complexity), the result that majority voting is optimal is very good to know.

The paper is written in a clear and understandable manner.

----------------- Cons:

I would have really championed the paper if these results were a lemma towards something bigger. For instance, I would have really liked if in addition, the paper contained error bounds for the Kendall tau case or at least some more insights towards "better" tie-breaking algorithms for the other distance metrics. I didn't quite understand parts of the experimental setup -- how are you fitting the Mallows model to the k-approval-votes data?

----------------- A couple of additional comments and suggestions:

The difference between k-approval voting and approval voting is not that subtle, and confusing one for the other can make a significant difference: it may be better to move the discussion on this distinction into the main text. Also please provide some justification as to why k-approval voting may be relevant for crowdsourcing applications that you mention.

Given that Theorem 2 is quite simple (and well known in folklore for plurality voting), the authors may want to impart a lower emphasis to it than their main results by relegating Theorem 2 to a proposition.

------------------ Post rebuttal:

The experiments are still quite fuzzy and its not clear if they are relevant. However, given that the primary focus of the paper seems to be the theoretical results, I am happy to completely ignore the experimental part. I was further really hoping that the rebuttal will contain a strong argument as to why k-approval voting is useful for crowdsourcing (or other applications affecting machine learning) as claimed in the introduction of the paper. However, I do not consider "k-approval is far more popular in practice than (unconstrained) approval voting simply because plurality is an instance of the former but not the latter" as a valid reason since this argument applies only to k=1 whereas the results of this paper are non-trivial only when k > 1. The "participatory budgeting" application does not seem relevant to the NIPS audience. I think this paper is much more suitable, and will be much more appreciated, in conferences such as AAAI or IJCAI.
Summary: The problem is important, the writing is clear, and the results are good and useful (particularly Theorem 1, part 2), and hence I am in favor of accepting the paper. However, I would have liked to see a further analysis of error guarantees or improved algorithms for me to champion the paper. Relevance to NIPS is questionable.

Submitted by Assigned_Reviewer_2

summary of the paper:

The paper presents a study of the optimality of approval voting under generative models for k-approval vote profiles using Mallows models. The analysis is motivated by the use of approval voting in crowdsourcing systems.

k-approval votes correspond to the case where people vote for exactly k candidates among n > k candidates. The winner(s) are the candidate(s) which received most votes. Under a specific generative model for the voting profiles (k-approval votes given by a set of people), approval voting is considered as optimal if the winners are exactly the same as given by the likelihood of the model, for all voting profiles. Mallows models are probability distributions over rankings (i.e. permutations), where the probability of a ranking is exponentially decaying as a function of its distance with a "center" ranking. 5 different models, corresponding to 5 different distances between rankings are studied. Given a ranking of candidates generated by the model, the corresponding k-approval vote is to take the k top-ranked candidates. The paper shows that for most distances, it is possible to find voting profiles such that the set of winners are not the same as those given by the maximum likelihood. The analysis is complemented by low (or high)-noise conditions. In most cases, the set of winners contain the best candidates as given by the likelihood model. The analysis is complemented by experiments on real data, showing that in practice the approval voting rule "is not that bad" compared to a non-parametric ranking model, but is much more tractable.

comments: the paper is clear and well-written. The study of approval voting is important and the question tackled is interesting. Moreover, the analysis is serious.

However, my feeling is that some gaps need to be filled: 1- motivation of k-approval voting and generative models over rankings: it seems that the common way to apply approval voting is to let people select less candidates if they are sufficiently confident. It does not seem possible to extend the analysis of the paper to this more usual setting, because the paper is based on ranking models that do not consider any kind of "confidence". Is there a way to extend the analysis to the usual approval voting scheme that I didn't see ? Or is k-approval voting better than standard approval voting in some situations ?

2- I would have found the analysis more complete with additional considerations of (1) the number of approval votes that are collected, and (2) the probability of failure. For instance, do we have that as the number of voting profiles grows, the probability of failure of approval voting tends to 0 for all these models ? I would easily believe that (but maybe I'm wrong). Then, what is the range of (k, n, number of votes) for which the non-optimality of approval voting is important ?

3- I found it somewhat strange in the eperiments not to use a Mallows model. It makes the experimental part somewhat independent of the analysis. It is interesting, however, to observe "some" suboptimality of approval voting compared to an MLE rule; however, the datasets are very small and it is unclear whether the difference is significant.

Summary: The question addressed in the paper is interesting. However, it is unclear for now if the analysis only describes a few corner cases or if the suboptimality of approval voting is an issue that should be considered in practice.

Submitted by Assigned_Reviewer_3

This is a nice paper about approval voting. While the results are not groundbreaking, they are novel, non-trivial, and a valuable research contribution given the prevalence of voting methods within crowdsourcing applications.

One note: It would have been nice if the authors showed that the example in section 3 is a counter-example for MD, CY, HM, and FR. This is the first part of theorem 1 but lacks actual proof in the paper (as far as I can tell). I understand there are space constraints but this should be in there.
Summary: This is a nice paper about approval voting. While the results are not groundbreaking, they are novel, non-trivial, and a valuable research contribution given the prevalence of voting methods within crowdsourcing applications.

Submitted by Assigned_Reviewer_4

This paper studies the optimality of "k-approval" voting aggregation using the approval voting rule (i.e., picking the item(s) with highest votes). In particular it studies optimality of this principle when the data is generated under the Mallows model (for different distance functions). The paper's main contributions are theoretical results which indicate that for most distance functions not all alternatives selected under approval voting are MLE optimal. However under the Kendall-Tau distance function (easily the most commonly used distance function with the Mallows model) they are equivalent.

Overall I found the paper to be an interesting read with intriguing results. That said, I'm uncertain if it reaches the high bar of acceptance for a NIPS submission. My main concerns are as follows:

- (Significance) I am unclear of the impact and significance of this work. What should a practitioner take away from this? What about other researchers working on this topic? This remains unclear to me despite reading the paper closely.

As the authors themselves state, approval voting isn't much worse than the MLE alternative in the situations where do they do not match. Furthermore the MLE estimator itself is infeasible to compute for large problems.

The secondary set of results (as presented in Sec. 4) aren't very impactful in my opinion given that they discuss (and conjecture) about optimality in extreme regions of the space, which most real problems do not fall under.

I would also have liked to see some discussion on why k-approval voting is preferable to approval voting as an feedback elicitation mechanism, given that the latter seems like the cognitively easier and more reliable of the two.

- (Originality) I would have also wanted to see some the connection made to the numerous results of approximation quality of rank aggregation under the Mallows model (such as the works by Nir Ailon, Cynthia Dwork and other researchers in the field). Given the decades of work on this problem, there is a rich body of literature to draw from, which can actually explain some of these results.

In general while this paper indicates non-optimality of some of these models, it does not indicate how suboptimal the approval voting solutions really are (which would have been a far more intriguing result).

- (Quality) In general the empirical results indicate only small differences in performance of the two methods. Given this I would have expected some kind of significance testing or error bar reporting as is standard practice.

- (Clarity) The paper was mostly clear. However there were notations and terminologies that were overloaded to the point of confusion. For instance the use of the d_{KT} notation on line 205 which is then followed by a completely different use of the same notation in the next two lines. I was also slightly confused at a few points in the paper with the use of the approval voting rule

terminology (which was designed for unconstrained approval feedback) was used under the k-approval voting problem.

In general much of the core ideas and insights are lost behind the sea of notation and consequently aren't made clear to the reader. Saying that non-optimality is due to some information being excluded is fine, but I would like to further understand what this information is and what are some preliminary ideas on feasible alternates that can capture this information.

A more minor comment: I also didn't understand the noise characterization for the datasets at the start of section 5. Just because differences are smaller does not make the underlying votes more noisy. Unless there were empirical results that indicated the same I wouldn't make that claim.
Summary: While the paper presents interesting results, its' impact is unclear.

Submitted by Assigned_Reviewer_5

The authors gave an elegant answer to an interesting question: suppose the rankings are generated from Mallows-like model equipped with various natural distances and each agent reports the set composed of top-k alternatives in the ranking, are the winners by approval voting the same as the alternatives with maximum posterior probability to be ranked in the top? The authors focused on five well-studied distance and obtained positive answer for Mallows. The authors also gave affirmative answers for neutral distance and 1-approval and (m-1)-approval votes.

I like the question asked in this paper especially because the positive answers imply that

it is easy to compute the alternatives with max posterior probability to be at the top (I think using the term MLE is misleading as the authors are indeed doing Bayesian inference w.r.t. 0-1 loss function). This is in sharp contrast to the hardness result for Mallows with Kendall Tau in [19]. The reason might be that because in this paper the sample space is different-it is composed of i.i.d. approval votes, so that some terms in the posterior distribution can be grouped.
Summary: This paper provides surprising and positive answers to an interesting new question about the important social choice problem.

Author Feedback
Author rebuttal: We thank all reviewers for their helpful feedback. To address the main concerns, in the paper we will extend the discussion of the differences between k-approval and (unconstrained) approval voting.

In response to the two concerns from the meta review:

1. k-approval is far more popular in practice than (unconstrained) approval voting simply because plurality is an instance of the former but not the latter. In the paper we also give an example of 8-approval used in EteRNA, and there are many other real-world examples, e.g., k-approval is used for so-called "participatory budgeting" by various cities in the US such as San Francisco, Chicago, and New York (see: http://goo.gl/AT3ERW).

2. While one cannot uniquely convert rankings to (unconstrained) approval votes, it is very natural to convert them to k-approval votes *for any fixed k* because a voter would simply report the k-prefix of his ranking when asked to report a favorite subset of candidates of a fixed size k.

Please find below our responses to other specific questions/concerns.

Assigned_Reviewer_1:

Our experiments did not use the Mallows model by design (we have given the matter careful thought). Instead of verifying our theoretical results, our goal was to check whether the conclusions we drew from our analysis of the Mallows model (near optimality of approval voting) hold true in real-world datasets.

Assigned_Reviewer_2:

We are closely familiar with the literature on rank aggregation under the Mallows model (e.g., results about approximating the Kemeny ranking, which is the MLE ranking). We are confident that this literature does not address our problem of finding the MLE *winner* given only *k-approval votes*. In this sense, this literature is more distantly related than the papers we cited about k-approval voting and its crowdsourcing applications. But we would be happy to briefly discuss the related work in this area as well.

On a technical level, k-approval cannot be transformed into a special case of Borda count: they are two different positional scoring rules (the former gives a weight of 1 to the first k candidates and 0 to the rest, while the latter gives a weight of m-i to the candidate in rank i).

The "smaller differences" were indeed positively correlated with higher noise in the data (according to various notions of noise) --- see [18] for a detailed analysis. We will make this explicit in the paper.

Assigned_Reviewer_4:

As you note, the counterexample is provided in the paper, but we promise to add to the online appendix the (rather tedious) calculation that supports it.

Assigned_Reviewer_7:

Error Bounds: We believe that by this you mean the sample complexity of approval voting (i.e., the error in pinpointing the best candidate as a function of the number of voters). Our paper follows a well-established approach of restricting attention to the structure of the MLE rule for a given noise model (see, e.g., [2,3,4,16,19,22]). Building on this approach, we focus on showing that the most popular way of aggregating k-approval votes may not be MLE, and study its connection to the MLE rule. But we agree that a compelling next step is to derive error bounds for approval voting and compare them with the bounds for the MLE (e.g., along the lines of the analysis in [8]) -- we will mention this in the paper.

Experiments: We are not fitting the Mallows model to the data. The MLE rule in the experiments is MLE for the empirically observed distribution of k-approval votes rather than for the distribution induced by the Mallows model. Thus, both the theory and the experiments study how approval voting compares with the MLE method for the distribution from which the votes are actually generated. The experiments test whether the qualitative predictions obtained through a theoretical analysis of the Mallows model also hold on real data where the noise doesn't necessarily follow the Mallows model. In practice, learning this distribution would be hard, but in our case it only amounts to learning 6 probabilities.

Theorem 2: Our main contribution in Theorem 2 is the exact conditions we pinpoint. For example, intuitively the veto rule should be optimal in any reasonable setting given (m-1)-approval votes. However, its condition is not satisfied under the Mallows model for natural distances like the Hamming distance and the Cayley distance -- a result that was initially counterintuitive to us.